# Factor in Fear: Interference Competition in Polymorphic Spadefoot Toad Tadpoles and Its Potential Role in Disruptive Selection

**DOI:** 10.3390/ani13071264

**Published:** 2023-04-06

**Authors:** Alexandru Strugariu, Ryan Andrew Martin

**Affiliations:** 1Department of Exact and Natural Sciences, Institute of Interdisciplinary Research, Alexandru Ioan Cuza University of Iași, 700057 Iași, Romania; alex.strugariu@gmail.com; 2Department of Biology, Case Western Reserve University, Cleveland, OH 44106, USA

**Keywords:** anuran, competition, disruptive selection, diversification, fear, phenotypic plasticity, resource polymorphism, specialization

## Abstract

**Simple Summary:**

Disruptive selection occurs when extreme phenotypes have a fitness advantage over intermediate phenotypes, thereby favoring the evolution and maintenance of diversity within populations. Resource competition within populations is an important cause of disruptive selection. Individuals can compete in two distinct ways: either by depleting resources (exploitative/indirect competition) or by impeding a competitor’s access to resources (interference/direct competition). However, it is generally assumed that exploitative competition is of greater importance for disruptive selection, while interference competition is rarely considered. Here, we experimentally explored the role of interference competition using a well-known example of disruptive selection, the Mexican spadefoot toad (*Spea multiplicata*), whose tadpoles develop into alternative resource-use specialists (omnivores and carnivores) while generalist, intermediate tadpoles are disfavored by disruptive selection. Our behavioral experiments revealed that intermediate tadpoles decreased their foraging in the presence of carnivores, while our competition experiment showed that interference competition with carnivores, but not omnivores, significantly lowered growth rates in intermediate tadpoles. Therefore, interference competition may be important in causing disruptive selection. Furthermore, we found that ‘fear’ (phenotypic responses to perceived predation risk) could mediate interference competition as carnivores (who are also cannibalistic) were responsible for both behavioral alterations and a decreased growth rate when direct interaction was possible.

**Abstract:**

Disruptive selection arises when extreme phenotypes have a fitness advantage compared to more-intermediate phenotypes. Theory and evidence suggest that intraspecific resource competition is a key driver of disruptive selection. However, while competition can be indirect (exploitative) or direct (interference), the role of interference competition in disruptive selection has not been tested, and most models of disruptive selection assume exploitative competition. We experimentally investigated whether the type of competition affects the outcome of competitive interactions using a system where disruptive selection is common: Mexican spadefoot toads (*Spea multiplicata*). *Spea* tadpoles develop into alternative resource-use phenotypes: carnivores, which consume fairy shrimp and other tadpoles, and omnivores, which feed on algae and detritus. Tadpoles intermediate in phenotype have low fitness when competition is intense, as they are outcompeted by the specialized tadpoles. Our experiments revealed that the presence of carnivores significantly decreased foraging behavior in intermediate tadpoles, and that intermediate tadpoles had significantly lower growth rates in interference competition treatments with carnivores but not with omnivores. Interference competition may therefore be important in driving disruptive selection. As carnivore tadpoles are also cannibalistic, the ‘fear’ effect may have a greater impact on intermediate tadpoles than exploitative competition alone, similarly to non-consumptive effects in predator–prey or intraguild relationships.

## 1. Introduction

Darwin [1] first postulated that natural selection would favor the evolution of trait divergence that minimizes competition between individuals, leading to the evolution of new phenotypes, and even new species. While evolutionary biologists following Darwin generally viewed this as a process occurring via interactions between different populations or species (i.e., ecological and reproductive character displacement; reviewed in [2]), intense competition for vital resources within populations can also lead to the evolution of intraspecific variation by disruptive selection [3,4,5,6]. Resource polymorphism—the occurrence of discrete intraspecific morphs with differential resource use within a population—is a potential evolutionary outcome of such disruptive selection [7]. Examples of resource polymorphism are plentiful in many taxa. Nevertheless, the level of divergence between morphs can vary tremendously between polymorphic species. In some cases, phenotypic differences (e.g., morphological, behavioral) can be very subtle and easily overlooked (e.g., pumpkinseed sunfish *Lepomis gibbosus*), while in other cases, differences are so dramatic that some morphs were initially described as separate species (e.g., Arctic char *Salvelinus alpinus*, tiger salamander *Ambystoma tigrinum*, spadefoot toads *Spea multiplicata,* and African finches *Pyrenestes ostrinus*) (reviewed in [7]).

Theory and empirical data suggest that intraspecific competition may result in disruptive selection when: (i) phenotypic variation is linked to resource use; (ii) competition is more intense between phenotypically similar individuals using the same limited resources; (iii) competition is density and frequency dependent; and (iv) underused resources are available (i.e., ecological opportunity). Thus, via negative frequency- and density-dependent selection, individuals with extreme resource-use traits that specialize in less common but underused resources exhibit a fitness advantage due to less intense competition with more common phenotypes [3,4,5,8]. Alternatively, it has also been argued that disruptive selection can also occur when resources are discrete (i.e., bimodal) and functional resource-use tradeoffs between them are strong (e.g., mobile and vulnerable vs. armored and sessile prey), resulting in higher fitness for specialists compared to generalists [9,10,11,12]. Importantly, these mechanisms can operate together in driving disruptive selection [5]. Although the role of intraspecific competition in disruptive selection and resource polymorphism is broadly acknowledged, competition can take several forms, a fact mostly disregarded in previous research [13,14]. In exploitative competition, individuals compete for shared, limited resources indirectly, while interference competition implies direct competition, involving antagonistic interactions that vary in intensity and consequences. Previous studies on disruptive selection have generally assumed competition to be exploitative or have not distinguished between the possible forms of competition, as exploitative competition is believed to be stronger than interference competition, particularly in less aggressive species that use more dispersed ecological resources [13,15]. Nevertheless, it has been recently suggested that interference competition may play a greater role in disruptive selection than previously considered, as it can have rapid effects on competing individuals and could modulate the negative frequency-dependent effects that arise from resource depletion and exploitative competition [13]. Interference competition may force less aggressive conspecifics to exploit underused resources, particularly with territorial species [16]. Furthermore, aggressive interactions should be more acute when extreme phenotypes not only compete with other phenotypes but also prey upon them [17]. Although cannibalism itself may not contribute greatly to disruptive selection (that said, see [18]), the perceived threat of being consumed or injured by a conspecific might have more powerful ecological and evolutionary consequences than previously thought.

Insights can be drawn from studies of predator–prey or intraguild relationships. Predator–prey relations have traditionally been viewed from the perspective of lethality (i.e., predator captures and consumes prey) and density-mediated interactions. However, theoretical and empirical evidence demonstrate that non-consumptive, trait-mediated interactions may have stronger effects on prey fitness than consumptive interactions [19,20]. Fear ecology theory thus predicts that the predator affects prey more than just through consumptive interactions, as prey species are under selection to avoid predators [19]. Indeed, in the presence of predators, prey species or beta-predators (in intraguild relationships) can exhibit considerable phenotypic (i.e., behavioral, physiological, developmental or morphological) alterations to reduce their risk of mortality [21,22,23]. However, these alterations reduce the prey’s overall fitness, as they carry considerable costs, either through the direct ‘cost’ of producing said alterations or through decreases in their reproductive investment and resource acquisition [24,25]. A meta-analysis conducted by Preisser et al. [20] showed that trait-mediated interactions are a major component in predator–prey relationships, comprising at least 50% of the total predator effects, by greatly increasing trophic cascades, and particularly in aquatic environments. While the factor of fear—phenotypic responses of potential prey in response to perceived predation risk—has received much attention in predator–prey and intraguild relations [20], its role in intraspecific resource competition, and hence in resource polymorphism and disruptive selection, has received little attention despite its potential importance.

Here, we experimentally investigated the effects of interference competition between resource-use morphs of the Mexican spadefoot toad (*Spea multiplicata*) tadpoles. *Spea* tadpoles plastically develop into alternative omnivore or a carnivore morphs in response to diet [26]. We focused on phenotypically intermediate tadpoles (in relation to the two morphs) that typically have low survival in natural ponds when resource competition is strong, as they are inferior competitors to both omnivore and carnivore morphs [5,14,27]. As intermediate tadpoles both compete with and are consumed by the carnivore morph tadpoles, and omnivores compete with intermediates and are consumed by the carnivores [5], the system mirrors an inter-specific intraguild relationship. We first ran behavioral trials where we asked if tadpoles would significantly alter their behavior (with regards to resource acquisition—foraging, swimming, or resting) in the presence of carnivore or omnivore tadpoles. We predicted that focal tadpoles would reduce foraging in the presence of carnivore competitors. Finally, we ran a competition experiment, allowing either exploitative or exploitative and interference competition for intermediate tadpoles with either omnivore morph or carnivore morph competitors. We predicted that interference competition would have the greatest effect during competition with carnivore morph competitors.

## 2. Materials and Methods

### 2.1. Study System

The Mexican spadefoot toad (*Spea multiplicata*) is a nocturnal, predominantly fossorial amphibian species that inhabits arid areas within the southwestern United States and Mexico. Adults of this species remain underground during the dry season, emerging to feed and breed during the summer monsoons. Breeding within a given ephemeral pond takes place on a single night, and development from egg to metamorphosis can occur in under two weeks [26,28]. Their tadpoles display a well-known form of resource polymorphism, as they develop into one of two environmentally induced ecomorphs: an ‘omnivore’ or ‘carnivore’ morph [26,28]. While the former morph is a dietary generalist that feeds predominantly on detritus and algae along with small zooplankton, the latter mostly consumes fairy shrimp, and sometimes other tadpoles, including conspecific ones [28,29]). The two morphs dramatically differ in a series of morphological traits, such as body shape, gut length, jaw muscle size, and mouthpart keratinization [28,30]. The distribution of phenotypes within a pond is determined by frequency-dependent disruptive selection driven by competition for resources [5,31]. While phenotypically intermediate tadpoles exist and can comprise the most common phenotype in some ponds, they have low survival in ponds with high densities of spadefoot tadpoles [5,14,32]. Disruptive selection disfavors intermediates as they are outcompeted by the two specialized ecomorphs [5]. This polymorphism is a plastic response, as the development of a tadpole into a carnivore requires an environmental cue, namely the ingestion of fairy shrimp or other tadpoles [26,28].

### 2.2. Behavioral Trials

To explore the potential behavioral responses of intermediate tadpoles to interference competition from carnivore tadpoles, we ran a series of experimental behavioral trials. We first collected live *S. multiplicate* tadpoles (~10 days old) by dip net from a temporary pond in Price Canyon, Arizona (‘Eagle’s Cry’) and transported them to the live animal holding facility at the Southwestern Research Station in Portal, Arizona. Prior to the experiment, we determined whether a tadpole was an omnivore morph, carnivore morph or intermediate by visual inspection of the jaw muscles, mouthparts and overall shape [5,26]. For each trial in the experiment, we introduced two intermediate tadpoles into a plastic tank (412.75 mm × 285.75 mm × 171.45 mm) filled with ~10,000 mL of aged water. We offered food in the tank, both in the form of fairy shrimp (*n* = 10) and ground dry fish food (detritus substitute), and allowed the tadpoles to acclimate for five minutes. We then recorded the tadpoles’ behavior every 30 s for five minutes and logged if they were swimming, feeding, or resting. Next, we added a carnivore tadpole into the tank and proceeded to record the intermediate tadpoles’ behavior for another five minutes, as previously mentioned. We then removed the carnivore tadpole, rinsed the tank and replaced the water in the tank, reintroduced the intermediate tadpoles, added food, and allowed them to acclimate for another five minutes. Subsequently, we introduced an omnivore tadpole into the tank and recorded the focal tadpoles’ behavior for five more minutes. We ran a total of 21 trials and focal tadpoles were not reused across trials.

### 2.3. Competition Experiment

We aimed to disentangle and measure the effects of exploitative competition alone from the combined effects of exploitative and interference competition using a microcosm laboratory experiment. For this experiment, we collected live *S. multiplicata* tadpoles (~7 days old) by dip net from a temporary pond near Portal, Arizona (‘Horseshoe’). We again transported them to the live animal holding facility at the Southwestern Research Station in Portal, Arizona. We measured the body mass with a digital balance to the nearest 0.001 g both prior to and after the experiment, and again determined whether a tadpole was an omnivore morph, carnivore morph, or intermediate as previously described [5,26]. On completion of the experiment, all tadpoles were euthanized by immersion in a buffered solution of MS-222 and preserved in 95% ethanol.

In order to disentangle the effects of purely exploitative (indirect) from exploitative and interference (direct) competition between the intermediate and specialized tadpoles, we measured growth as a proxy for fitness during a 10-day experiment, in which intermediate tadpoles competed for food against either omnivore or carnivore tadpoles, in both mixed (interference + exploitative) or single (exploitative) treatments, as follows:

To simulate the conditions of both exploitative and interference competition acting together, we reared intermediate tadpoles (*n =* 32), together with either a carnivore (*n =* 16) or an omnivore (*n =* 16), for the entire duration of the experiment. We housed the tadpole pairs of both treatments in plastic tanks (343 mm × 209.5 mm × 120.65 mm) filled with 3250 mL of water. As spadefoot toad tadpoles are known to accelerate development in rapidly evaporating ponds [33], we inspected the water levels daily and topped up when needed in order to maintain the initial volume throughout the experiment. Carnivore treatment tadpole pairs were fed 30 fairy shrimp every 24 h, while the omnivore treatment pairs were fed 20 mg of ground, dry cichlid pellets (as a substitute for detritus) every 48 h. We fed the omnivore treatment pairs less frequently, as the consumption of the cichlid food is slower than that of fairy shrimp. Consequently, tadpoles within this treatment competed for food while being exposed to potential antagonistic interactions and/or fear-induced behavioral changes.

For simulating purely exploitative competition, we reared intermediate (*n =* 32), carnivore (*n =* 16), and omnivore (*n =* 16) tadpoles alone in individual plastic tanks. We assigned each intermediate tadpole a specific carnivore (carnivore competition treatment; *n =* 16), or an omnivore (omnivore competition treatment; *n =* 16) tadpole competitor with which they competed for food in an exploitative manner. Specifically, carnivore treatment intermediates and carnivores were each fed 15 fairy shrimp every 24 h, while omnivore treatment intermediates and omnivore tadpoles were fed 10 mg of dry cichlid food every 48 h. After the introduction of food, tadpoles were allowed to forage for one hour, after which each tadpole was removed from its tank and introduced in its competitor’s tank, thus having access to the resources remaining after one hour of competitor foraging. Tadpoles thus competed for the same resources without experiencing direct interaction. For this treatment, we filled the tanks with 1625 mL of water, representing half of the volume used in the interference treatment, as only one tadpole was present at any given time in a tank, as opposed to two in the interference treatment. The water level was inspected and maintained constant, as with the previous treatment. It is important to note that while this design minimized interference competition, it could not remove all potential sources of interference competition. There was no possibility for direct, physical, or visual interactions between competing tadpoles; however, we could not eliminate the possibility of interference competition via chemical interactions.

### 2.4. Data Analysis

*Behavioral trials:* To analyze the data from our behavioral trials, we fit separate generalized linear mixed models (GLMM) with a binomial error structure for each of the three measured behaviors: foraging, swimming, or resting (coded as ‘1’ if a behavior was observed at each timepoint and ‘0’ if not). Each model included treatment (control, carnivore added, omnivore added) as a fixed effect, and the focal pair ID as a random effect. The mixed models were fit with the *glmer* function from the {lme4} library [34] in R (version 3.6.1; [35]). We used analysis of deviance to assess the statistical significance of treatment. We ran post hoc tests using the *emmeans* function from the {emmeans} library [36] to evaluate the pairwise treatment comparisons.

*Competition experiment*: To compare the effects of purely exploitative (indirect) from the combined effects of exploitative and interference (direct) competition between the intermediate and specialized tadpoles, we fit a linear model using the *lm* function in R with the difference in growth of the focal (intermediate morph) tadpoles’ mass minus the growth of their competitor as the response variable. We included competition treatment, competitor type, and their interaction as predictors. We used *F*-ratios and calculated Type 3 sums of squares to assess the statistical significance of these effects. The *emmeans* function was again used to evaluate the pairwise treatment by competitor type comparisons.

## 3. Results

### 3.1. Behavioral Trials

We used a behavioral trial to examine how tadpoles intermediate in phenotype would respond to omnivore or carnivore morph tadpoles. Introducing a third tadpole competitor significantly changed the frequency of foraging (χ^2^ = 40.86, *p* < 0.0001), swimming (χ^2^ = 7.16, *p* = 0.0278), and resting behaviors (χ^2^ = 15.75, *p* = 0.0003). Post hoc tests showed that focal tadpoles decreased foraging in the presence of an additional competitor and that this effect was greatest for carnivore competitors (Table 1, Figure 1). In contrast, swimming behavior increased in the presence of carnivores, and only when compared to the control but not the omnivore treatment (Table 1, Figure 1). Finally, the focal tadpoles more frequently rested at the bottom of the tank after the introduction of a second competitor, but this did not differ between omnivore and carnivore competitors (Table 1, Figure 1).

### 3.2. Competition Experiment

We aimed to disentangle the effects of interference from exploitative competition using a laboratory microcosm experiment. We found a significant interaction between the competition treatment and competitor type on the growth of our focal intermediate tadpoles (competition treatment: *F*_1,60_ = 40.33, *p* < 0.0001; competitor type: *F*_1,60_
*=* 0.00, *p* = 0.999; treatment x type: *F*_1,60_
*=* 28.14, *p* < 0.0001). Post hoc tests revealed that the focal tadpoles grew significantly worse than their carnivore competitors when both exploitative and direct interference competition were possible, but that the effects of competition did not differ among the other treatment combinations (Table 2, Figure 2).

## 4. Discussion

Resource competition is an important cause of evolutionary diversification [6,37]. Fitness declines when individuals of the same or separate species compete for shared, limited resources. Individuals who exploit alternative resources, however, can escape competition and gain a fitness benefit, driving divergent selection and potentially, evolutionary diversification within and between species [37,38]. Most empirical and theoretical research into this process, however, focuses, often implicitly, on the role of exploitative competition. Consequently, how interference competition affects competitively driven disruptive selection and ecological diversification is not well understood [13,14]. We addressed this question in polymorphic spadefoot toad tadpoles (*S. multiplicata*). First, using a behavioral experiment, we found that the focal phenotypically intermediate tadpoles foraged less when a third tadpole was introduced, and that the introduction of carnivore morph tadpoles caused the greatest decrease in foraging (Figure 1, Table 1). By experimentally manipulating the forms of competition tadpoles experienced, we next found that the focal tadpoles grew more slowly than carnivore morph competitors, but only when they were allowed to interact directly (e.g., when both exploitative and interference competition was possible) compared to when the interference competition was minimized. Together, these results suggest that interference competition plays an important role in the evolution and maintenance of resource polymorphism in spadefoot toad tadpoles, and perhaps in other systems as well.

Why was competition more intense when our focal tadpoles directly interacted with carnivore competitors (Figure 2, Table 2)? The responses to carnivores in the behavioral experiment point to indirect effects of fear leading to decreased foraging behavior (Figure 1, Table 1). While there was no direct predation within our competition experiment, carnivore morph tadpoles do engage in cannibalism in nature and in experimental settings [39]. We cannot, however, rule out that the difference in behavioral responses to the introduction of carnivore versus omnivore competitors could instead be attributed to our experimental design. As the carnivore competitor was always introduced first in the behavioral experiment, the responses to omnivores could have been influenced by the prior introduction. In both of our current experiments, it seems possible that focal tadpoles were responding to visual and/or tactile cues of competitors. Tadpoles in other species respond to these cues in response to both predators and conspecifics [40,41,42,43,44]. While tadpoles [41], including *Spea* tadpoles [29,39,45,46], also respond to chemical cues, it is unlikely that such cues mediated the responses we found in our experiments since both treatments were exposed to potential chemical cues.

Results from previous studies also suggest that interference and exploitative competition act together in *S. multiplicata* tadpoles. *Spea multiplicata* tadpoles exhibit kin recognition and can act altruistically towards kin [29,39,45,47,48]. In a competition experiment manipulating both relatedness and resource availability, tadpoles competed less with siblings than non-siblings when reared together. However, this only occurred when alternative resources were available to them [49], suggesting that unrelated competitors were engaging in interference competition while siblings avoided doing so. It is perhaps surprising then that exploitative competition had little effect on its own in the current competition experiment. Previous studies have found that exploitative competition is generally strong among spadefoot toad tadpoles [5,14,27,31]. Potentially, we found little effect of exploitative competition here either because resources were not limiting in the experiment or because we did not allow individual competitors enough time to exploit their conspecifics’ resources within the exploitative competition treatment.

Fear responses and other forms of interference competition could allow some individuals to monopolize profitable resources and force others to use less preferred alternatives [16]. For *Spea* tadpoles, interference competition could enable carnivores to discourage competitors from using the more profitable shrimp resource [14,50], in addition to the advantages carnivores already possess in exploitative competition for fairy shrimp [5]. Because the distribution of fairy shrimp within ponds is clumped, carnivore morph tadpoles can behaviorally monopolize this resource, even when carnivores are at relatively low frequency in the population. This in turn could increase competition among less carnivore-like tadpoles for alternative resources—for which phenotypically intermediate tadpoles are poorer competitors compared with omnivore morph tadpoles—and strengthen disruptive selection [14]. Indeed, disruptive selection is both widespread and of strong magnitude in *S. multiplicata* populations [32]. Fear as a behavioral mediator of interference competition might be expected to be common within resource polymorphisms where cannibalism occurs between morphs; however, this possibility has not been widely explored [14,18].

Rather than the fear of predation, how might cannibalism itself affect competition, disruptive selection, and the evolution of resource polymorphism? There are several possibilities. First, by consuming other conspecifics, cannibalism could weaken exploitative competition for resources [18,51,52,53,54]. For example, *Spea* carnivores tend to target smaller, omnivore morph tadpoles [28,55]. As a consequence, cannibalism in this system might be predicted to reduce competition by providing an additional resource for carnivores and eliminating omnivore competitors. However, by reducing the range of resource-use phenotypes in the population by targeting omnivores, cannibalism might instead increase negative frequency-dependent competition and thereby strengthen disruptive selection [5]. Moreover, within fish planktivore–benthivore resource polymorphisms, cannibalism between age cohorts can stabilize consumer–resource dynamics in populations, thereby promoting the evolution and maintenance of resource polymorphism [18,53,54].

While the role of interference competition in disruptive selection or resource polymorphism is rarely investigated, there is evidence of its potential importance from other systems. For example, the determinants of resource specialization are complex in the oystercatcher, *Haematopus ostralegus* [56]. Individuals specialize in different prey items (e.g., mussels vs. worms) and use different feeding strategies to specialize within a given prey (e.g., hammering vs. stabbing mussels) [56,57]. Both learning and beak morphology/beak wear contribute to the choice of resource use in oystercatchers [56], yet young and subordinate individuals can be excluded from their preferred, profitable diets by more dominant individuals [58].

## 5. Conclusions

If the role of interference competition has been understudied for its potential role in causing disruptive selection in resource use, other areas of research suggest that interference competition could be important in the evolution and maintenance of diversity. Within species, interference competition can, in part, drive fitness tradeoffs between alternative male reproductive strategies, resulting in negative frequency-dependent selection and the maintenance of multiple strategies [59]. Between species, competition for mates, territories, or other aggressive interspecific interactions can lead to evolutionary divergence and coexistence (i.e., character displacement) [2,60,61]. For plants and microbes inhabiting patchy habitats, interference competition can promote greater species diversity [62,63,64,65]. An emergent outcome of these studies and our own suggests that interference competition may be an important, if understudied, mechanism in the evolution and maintenance of inter- and intra-specific variation.

## Figures and Tables

**Figure 1 animals-13-01264-f001:**
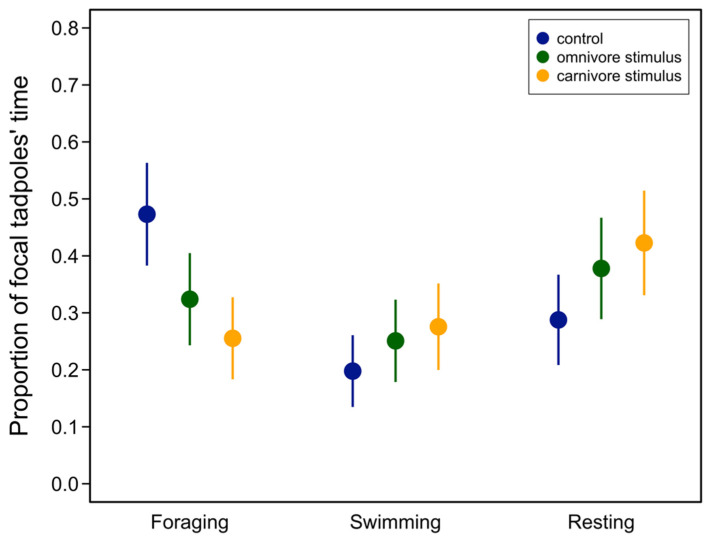
Proportion of focal intermediate tadpoles’ behaviors after introduction of an additional omnivore or carnivore competitor. Filled symbols represent mean proportion for each behavior and vertical lines represent 95% confidence intervals from the fitted model. See Table 1 for contrasts.

**Figure 2 animals-13-01264-f002:**
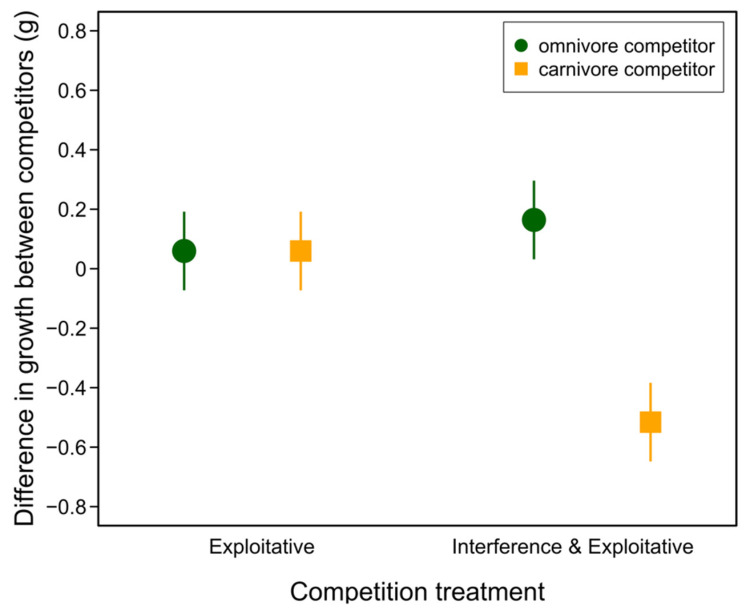
Differences in growth rate between intermediate morph focal tadpoles and their competitor. Filled symbols represent predicted mean growth and vertical lines represent 95% confidence intervals from the fitted model. See Table 2 for contrasts.

**Table 1 animals-13-01264-t001:** Post hoc analyses of the effect of competitor type on tadpole behavior are provided including estimates of the contrast, their standard errors and test statistics and *p*-values for the pairwise differences between factor levels. Statistically significant *p*-values at the 0.05 level are indicated in bold font. *p*-values were adjusted using the false discovery rate for multiple tests. Values are presented to three significant digits.

Behavior	Contrast	Estimate	SE	Z	*p*
Foraging	carnivore–control	−0.217	0.0349	−6.24	**<0.0001**
	carnivore–omnivore	−0.0685	0.0324	−2.11	**0.0347**
	control–omnivore	−0.149	0.0352	4.23	**<0.0001**
Swimming	carnivore–control	0.0778	0.300	2.58	**0.0289**
	carnivore–omnivore	0.0247	0.309	0.801	0.423
	control–omnivore	−0.0531	0.029	−1.82	0.102
Resting	carnivore–control	0.135	0.0346	3.89	**0.003**
	carnivore–omnivore	0.0448	0.0355	1.26	0.206
	control–omnivore	−0.0902	0.0339	−2.65	**0.0118**

**Table 2 animals-13-01264-t002:** Post hoc analyses of the interaction between competitor type and competition treatment are provided including estimates of the contrast, their standard errors and test statistics, and *p*-values for the pairwise differences between factor levels. Statistically significant *p*-values at the 0.05 level are indicated in bold font. *p*-values were adjusted using the false discovery rate for multiple tests. Values are presented to three significant digits.

Contrast	Estimate	SE	T Ratio	*p*
e,c—i,c	0.575	0.0906	6.35	**<0.0001**
e,c—e,o	−0.0000625	0.0906	−0.001	0.999
e,c—i,o	−0.104	0.0906	−1.15	0.305
i,c—e,o	−0.575	0.0906	−6.35	**<0.0001**
i,c—i,o	−0.679	0.0906	−7.50	**<0.0001**
e,o—i,o	−0.104	0.0906	−1.15	0.304

e = exploitative treatment; i = interference & exploitative treatment; c = carnivore competitor; o = omnivore competitor.

## Data Availability

Data are publicly archived on The Open Science Framework, doi: 10.17605/OSF.IO/BNM4H.

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
