# Peer review of "Factor in Fear: Interference Competition in Polymorphic Spadefoot Toad Tadpoles and Its Potential Role in Disruptive Selection"

_animals, 2023, doi:10.3390/ani13071264_

Round 1
Reviewer 1 Report
This is a well written paper reporting the results of a few small laboratory experiments. In the interpretation of these results, some caveats might warrant consideration.
In the behavioral trials, always the carnivore competitor was added before the omnivore. Hence, the slightly stronger reduction in foraging time might as well be interpreted as stronger effect of the first disturbance during the trials, compared to the second one. If "fear" plays a role in interference competition, why should the focal tadpoles be afraid of an omnivore competitor (effects of the third competitor added are similar in both cases; fig. 1)?
Spea tadpoles develop in two alternative morphs, with intermediate individuals being present. These intermediates are no distinct group, but occupy the morphospace between the two named morphs. For these experiments, tadpoles were categorized by visual inspection, but although they were preserved after the competition experiment, no morphological measurements are reported to verify their position in morphospace.
The conclusion that "visual or tactile cues" may be involved in interference competition thus is not strongly supported by the evidence presented in this manuscript. The results are worth publishing, to draw attention to an understudied aspect of phenotypic plasticity, but in view of the extent and quality of the data a Short Note or Communication may be a more appropriate format than a full Article.
Detailed comments:
line 2: Questions in titles are questionable on principle, and this question is particularly puzzling as readers may regard "Factor" as a noun (translation programs do it). If you recognize "factor in" as a verb, it remains unclear who should do this, to whom this question is addressed: to the readers, researchers, tadpoles? Suggest to rephrase the title.
line 91: better "prey on them" (than: predate upon them)
line 135: "They ..."; unclear reference, as in the previous sentence "the ... species" is singular. Suggestion: "Adults of this species ..."
line 337: "Fear as a form of interference competition" ?? Behaviors inducing fear in competitors can be a form of competition, but to equate fear with competition seems to confound categories.
Author Response
Reviewer 1:
This is a well written paper reporting the results of a few small laboratory experiments. In the interpretation of these results, some caveats might warrant consideration.
Authors' response: Thank you for your helpful review. We detail our edits and responses below.
In the behavioral trials, always the carnivore competitor was added before the omnivore. Hence, the slightly stronger reduction in foraging time might as well be interpreted as stronger effect of the first disturbance during the trials, compared to the second one. If "fear" plays a role in interference competition, why should the focal tadpoles be afraid of an omnivore competitor (effects of the third competitor added are similar in both cases; fig. 1)?
Authors' response: We agree that the experimental design makes it difficult to interpret the effect of the omnivore introduction, independent from the carnivore introduction. Tadpoles could have been habituated to the second introduction, as the reviewer suggests. Alternatively, however the focal tadpoles may have responded more strongly to the omnivore introduction in reponse to the earlier introduction. In response to the reviewers comment, we have added the following sentence to the Discussion, to acknowledge this caveat to the behavioral experiment:
(Line 320) "We cannot however rule out that the difference in behavioral responses to the introduc-tion of carnivore versus omnivore competitors could instead be attributed to our experimental design. As the carnivore competitor was always introduced first in the behavioral experiment, the responses to omnivores could have been influenced by the prior introduction."
Spea tadpoles develop in two alternative morphs, with intermediate individuals being present. These intermediates are no distinct group, but occupy the morphospace between the two named morphs. For these experiments, tadpoles were categorized by visual inspection, but although they were preserved after the competition experiment, no morphological measurements are reported to verify their position in morphospace.
Authors' response: While many spea pond populations exhibit a bimodal distribution of resource-use phenotypes, other are unimodal with tadpoles of intermediate phenotype predominating (populations can also differ in mean phenotype). It is correct that tadpole morphotype was determined by visual inspection, as has been previously done in other published studies for this system.
The conclusion that "visual or tactile cues" may be involved in interference competition thus is not strongly supported by the evidence presented in this manuscript.
Authors' Response: We agree and we intended the original statement to present how interference competition may be mediated in this system. We have edited this sentence to better convey that. We now write: "In both of our current experiments, it seems possible that focal tadpoles were responding to visual and/or tactile cues of competitors"
The results are worth publishing, to draw attention to an understudied aspect of phenotypic plasticity, but in view of the extent and quality of the data a Short Note or Communication may be a more appropriate format than a full Article.
Authors' Response: Thank you for your review and helpful comments. We have not taken this specific suggestion, as we feel our results are best explained in the current format.
Detailed comments:
line 2: Questions in titles are questionable on principle, and this question is particularly puzzling as readers may regard "Factor" as a noun (translation programs do it). If you recognize "factor in" as a verb, it remains unclear who should do this, to whom this question is addressed: to the readers, researchers, tadpoles? Suggest to rephrase the title.
Authors' Response: We have edited the title, as suggested.
line 91: better "prey on them" (than: predate upon them)
Authors' Response: done
line 135: "They ..."; unclear reference, as in the previous sentence "the ... species" is singular. Suggestion: "Adults of this species ..."
Authors' Response: done
line 337: "Fear as a form of interference competition" ?? Behaviors inducing fear in competitors can be a form of competition, but to equate fear with competition seems to confound categories.
Authors' Response: we have edited this sentence. We now write: "Fear as a behavioral mediator of interference competition might be expected to be common within resource polymorphisms where cannibalism occurs between morphs, however this possibility has not been widely explored."
Reviewer 2 Report
Dear authors, I really enjoyed reading your ms and applause for the choice of the animal model and experimental design. I can hardly imagine a better-suited tadpole-system for these kind of experiments. Analyses and results are straightforward and well-explained, nothing to critisize. My only (small) issue with the ms is that you do not explain in detail the meaning of fear in this context. A few more issues are marked in the commented pdf. My only formal issue is with the two types of abstract/summary, which are virtually the same. This illustrates again that an abstract alone would be sufficient - I do not understand the intention of "Animals" asking the authors for both.
Congratulation, really a fine study worth to enter in pertinent textbooks.
Ulrich Sinsch

Author Response
Reviewer 2:
Dear authors, I really enjoyed reading your ms and applause for the choice of the animal model and experimental design. I can hardly imagine a better-suited tadpole-system for these kind of experiments. Analyses and results are straightforward and well-explained, nothing to critisize. My only (small) issue with the ms is that you do not explain in detail the meaning of fear in this context. A few more issues are marked in the commented pdf. My only formal issue is with the two types of abstract/summary, which are virtually the same. This illustrates again that an abstract alone would be sufficient - I do not understand the intention of "Animals" asking the authors for both.
Congratulation, really a fine study worth to enter in pertinent textbooks.
Authors' response: Thank you for your review and suggestions. We have taken all of the suggestion in the marked pdf, with the single exception of retaining '10-day' rather than '10-days' as we believe the former is correct in this context. In addition, we now define our meaning of 'fear' in this context (line 118). We have also slightly modified the simple summary to better differentiate it from the abstract and to provide a definition of 'fear' at the earliest possibility.